# Positron Range Corrections and Denoising Techniques for Gallium-68 PET Imaging: A Literature Review

**DOI:** 10.3390/diagnostics12102335

**Published:** 2022-09-27

**Authors:** Prodromos Gavriilidis, Michel Koole, Salvatore Annunziata, Felix M. Mottaghy, Roel Wierts

**Affiliations:** 1Department of Radiology and Nuclear Medicine, Maastricht University Medical Center, 6229 HX Maastricht, The Netherlands; 2School for Oncology and Reproduction (GROW), Maastricht University, 6200 MD Maastricht, The Netherlands; 3Nuclear Medicine and Molecular Imaging, Katholieke Universiteit Leuven, 3000 Leuven, Belgium; 4Unit of Nuclear Medicine, TracerGLab, Department of Radiology, Radiotherapy and Hematology, Fondazione Policlinico Universitario A. Gemelli IRCCS, 00168 Rome, Italy; 5Department of Nuclear Medicine, RWTH University Hospital, D-52074 Aachen, Germany

**Keywords:** positron emission tomography, Gallium-68, positron range, spatial resolution, noise, low count

## Abstract

Gallium-68 (^68^Ga) is characterized by relatively high positron energy compared to Fluorine-18 (^18^F), causing substantial image quality degradation. Furthermore, the presence of statistical noise can further degrade image quality. The aim of this literature review is to identify the recently developed positron range correction techniques for ^68^Ga, as well as noise reduction methods to enhance the image quality of low count ^68^Ga PET imaging. The search engines PubMed and Scopus were employed, and we limited our research to published results from January 2010 until 1 August 2022. Positron range correction was achieved by using either deblurring or deep learning approaches. The proposed techniques improved the image quality and, in some cases, achieved an image quality comparable to ^18^F PET. However, none of these techniques was validated in clinical studies. PET denoising for ^68^Ga-labeled radiotracers was reported using either reconstruction-based techniques or deep learning approaches. It was demonstrated that both approaches can substantially enhance the image quality by reducing the noise levels of low count ^68^Ga PET imaging. The combination of ^68^Ga-specific positron range correction techniques and image denoising approaches may enable the application of low-count, high-quality ^68^Ga PET imaging in a clinical setting.

## 1. Introduction

In recent years, ^68^Ga-labeled radiotracers have been increasingly used in clinical positron emission tomography (PET). The commercial availability of ^68^Ge/^68^Ga generators facilitates the in-house production of ^68^Ga-labeled radiotracers, with applications in oncological, cardiovascular, infection, and inflammation PET imaging [1,2,3,4]. Particularly, ^68^Ga-labeled somatostatin receptor PET imaging is extensively used for neuroendocrine tumor imaging while the ^68^Ga-labeled prostate-specific membrane antigen (PSMA) has been shown to provide valuable information for the clinical management of prostate cancer patients.

Compared to the most used PET radionuclide ^18^F, ^68^Ga emits high-energy positrons (^68^Ga: *E*_max_ = 1.899 MeV; ^18^F: *E*_max_ = 0.635 MeV) [5]. Consequently, ^68^Ga exhibits an increased positron range, which is the travelled distance between the location of the decaying parent nucleus and the location of positron annihilation. Therefore, the larger the positron range, the more the measured PET signal can become blurred. This degradation of the spatial resolution of PET images depends on the underlying tissue type with a more severe degradation for lower-density tissues [6,7,8,9,10]. Studies have confirmed a negative impact on the spatial resolution for radionuclides emitting high-energy positrons, while phantom experiment revealed lower recovery coefficients (RC) and a reduced quantitative accuracy for radionuclides emitting high-energy positrons [6,8,9,10,11,12].

As good manufacturing practice (GMP)-compliant ^68^Ge/^68^Ga generators and labeling kits are generally expensive, it is essential to maximize the cost effectiveness of ^68^Ga PET examinations. Therefore, in clinical practice, the reduction in both the administered activity and PET acquisition time is highly relevant to maximize patient throughput. Additionally, it is important from radiation exposure and patient comfort perspectives. However, this will result in PET acquisitions with lower count statistics and thus increased statistical noise. Along with spatial resolution, noise is another prominent factor that can degrade image quality, lesion detectability, and quantitative accuracy of PET images. In recent years, deep learning approaches have been successfully applied for denoising PET images acquired with lower administered activities or shorter acquisition times [13,14,15].

In the current paper, we aim to provide a literature review of recently developed software-based techniques to enhance the PET image quality and quantification by compensating for the positron range effect and reducing the noise, specifically for ^68^Ga-labeled radiotracers. The abbreviations used in this paper are summarized in Table 1.

## 2. Materials and Methods

A literature review was conducted using the search engines PubMed and Scopus. Different queries were created for the positron range correction techniques and denoising approaches. For the positron range correction, we created queries incorporating terms related to PET, ^68^Ga, and positron range, while for noise reduction, terms such as PET, ^68^Ga, noise, or low count were used. The queries of the positron range corrections techniques are presented in Table A1 and Table A2 in Appendix A for PubMed and Scopus, respectively. Table A3 and Table A4 in Appendix A describe the queries that were used to retrieve articles related to noise reduction from PubMed and Scopus, respectively. As we aim to provide an overview of recent literature, only articles published between January 2010 and 1 August 2022 were included. From the retrieved records, duplicates were removed, then title and abstract screening was performed. Lastly, full-text screening was performed. In Figure 1, the chart of the screening process is illustrated.

## 3. Results

The total number of research articles generated from the search queries was 1162, with 126 publications related to positron range correction and 1036 publications related to denoising. After the removal of duplicates and title and abstract screening, 38 articles in total remained and were screened based on the full-text. During this process, another 14 publications were excluded (*n* = 2: Positron range correction techniques, which have already been described in other included articles, *n* = 9: No noise reduction technique, *n* = 1: Not studied for ^68^Ga, *n* = 1: Not a novel technique, *n* = 1: No noise-related metric used for the analysis, where the term *n* depicts, in this case, the number of full-text articles excluded based on the corresponding reason). The total number of articles included in this literature review was 24, with 8 articles describing a positron range correction technique and 16 articles introducing a novel denoising technique. Figure 1 presents the workflow and results of the literature screening.

Four different categories of techniques were identified, consisting of articles reporting on reconstruction-based positron range correction (*n* = 5), post-reconstruction positron range correction (*n* = 3), reconstruction-based noise reduction (*n* = 11), and deep learning noise reduction (*n* = 5), as shown in Figure 2. First, positron range correction techniques will be described followed by image noise reduction techniques.

### 3.1. Positron Range Correction Techniques

Based on the included literature, a variety of approaches have recently been developed for ^68^Ga-specific positron range correction. These can be grouped into two approaches: (1) Incorporating positron range models in the reconstruction algorithm (reconstruction-based correction, *n* = 5) and (2) approaches applying post-reconstruction positron range correction (post-reconstruction correction, *n* = 3), as seen in Figure 2. Reconstruction-based corrections generally used blurring kernels to model the positron range during the iterative reconstruction process [16,17,18,19,20], while post-reconstruction corrections applied deblurring kernels or deep learning techniques on the reconstructed images [21,22,23].

#### 3.1.1. Reconstruction-Based Correction

The reconstruction-based techniques correct for the positron range by incorporating positron range distribution models in the image reconstruction algorithm. In contrast to system response modeling techniques, in which position-dependent response modeling is performed based on scanner characteristics, positron range modeling typically requires patient-specific modeling depending on the underlying patient-specific tissue characteristics [6,7,8,9,10]. The reconstruction-based positron range corrections can be divided into three subcategories: Tissue-independent [16,17,18], homogeneous tissue-dependent [16,17,18,19], and heterogeneous tissue-dependent corrections [16,19,20]. Those corrections are applied in the forward projection step of the reconstruction algorithm, while a simplified positron range modeling is introduced in the back projection step, unless it is stated otherwise.

##### Homogeneous Tissue-Dependent Positron Range Correction

In clinical and preclinical practice, the scanned object comprises several different tissue types. As the positron range effect varies for different tissue types [6,7,8,9], a tissue-independent correction, as described in the previous section, is an oversimplification of the real situation. Therefore, techniques employing different blurring kernels based on different homogeneous tissues were developed [16,17,18,19]. Similar to the tissue-independent approach, the location of the decaying parent nucleus, the travelled distance of the positron, and the location of positron annihilation were assumed to occur in the same homogenous tissue; however, depending on the tissue (water, soft, lung, or bone tissue), different kernels were applied. For each tissue of interest, positron range distribution profiles of ^68^Ga were generated and used to create isotropic blurring kernels. For each voxel, a specific blurring kernel was applied depending on the respective tissue type of that voxel. The positron range distribution was obtained from Monte Carlo simulations or analytically [16,17,18,19].

Using phantom simulations, Bertolli et al. demonstrated that the activity recovery of the homogeneous tissue-dependent approach in lung tissue was approximately 90% of the simulated activity [17]. Compared to the non-corrected images, the homogeneous tissue-dependent approach achieved greater than double recovered activity for both lung and bone tissues [17,18]. Additionally, RC up to approximately 80% was obtained from reconstruction implementing homogeneous tissue-dependent correction, while the reconstruction without positron range correction achieved RC up to approximately 40% [18]. The tissue-dependent positron range correction also improved the full width half maximum (FWHM) up to 28% [18] and the full width tenth maximum (FWTM) up to 50% compared to the non-corrected reconstructions [16]. On top of that, PET data from mice injected with ^68^Ga-DOTATOC in a microPET/CT system revealed an increment in the tumor-to-background ratio (TBR) of approximately 80% compared to reconstruction without any correction [16]. Aside from that, it was demonstrated that implementation of the positron range kernels in both the forward- and backward projection steps resulted in similar image quality compared to the implementation of the kernel in only the forward projection step [18]. However, the first approach introduced a smoothing effect and reduced the convergence speed [18]. Simulating a PET/MRI scanner, Kraus et al. demonstrated that the tissue-dependent approach improved the image contrast up to approximately 50% and reduced the misplaced activity concentration in soft tissue compared to the reconstruction without positron range correction [19]. However, application of the tissue-dependent correction introduced Gibbs artifacts, noise was increased due to deblurring and even doubled in some cases, while the reconstruction time can also be doubled [16,17]. To achieve the same image noise between reconstructions with and without homogeneous tissue-dependent positron range correction, Cal-González et al. increased the number of image updates of the reconstruction without correction by a factor of 2.6 [16].

##### Heterogeneous Tissue-Dependent Positron Range Correction

It is possible that the decay of the parent nucleus and positron annihilation does not occur in the same tissue, meaning that the positron travels through different tissue types with different densities. Therefore, kernels assuming homogeneous tissues are only an approximation of the underlying situation. Consequently, more realistic heterogeneous tissue-dependent positron range corrections were developed [16,19,20].

Kraus et al. used Monte Carlo simulations to obtain the positron range distribution profiles for heterogeneous tissue configurations with different tissue borders [19]. Those simulated positron range data were used to create heterogeneous blurring kernels. This approach enhanced the image contrast to greater than 55% [19]. It further and further reduced the misplaced activity concentration in soft tissue compared to the homogeneous tissue approach [19]. Cal-González et al. also introduced a heterogeneous, tissue-dependent approach for PET/CT, in which the positron range profiles were calculated analytically [16]. Phantom simulations revealed an improvement of the FWTM up to 50% compared to the non-corrected images, similar to the homogeneous correction [16]. However, microPET imaging of mice injected with ^68^Ga-DOTATOC demonstrated that the heterogeneous correction increased TBR by more than 90% compared to reconstruction without correction [16]. Additionally, it produced better results in regions along tissue borders compared to homogeneous tissue-dependent techniques [16]. Another correction was proposed by Kertész et al. [20]. They merged parts of homogeneous tissue-dependent kernels to create a positron range kernel for heterogeneous tissues. The homogeneous tissue-dependent kernels were based on distribution profiles extracted from Monte Carlo simulations. Phantom measurements showed that the proposed correction resulted in an improvement of the RC by up to 33%, contrast recovery (CR) by up to 49%, and in some cases, more than doubled the contrast-to-noise ratio (CNR) compared to the non-corrected reconstruction [20].

Reported drawbacks of heterogeneous tissue-dependent correction techniques were Gibbs artifacts and increased reconstruction times [16,20]. Cal-González et al. observed a ninefold increment of the reconstruction time compared to the reconstruction without correction for the same system without any parallel implementation [16]. For a parallel implementation, the reconstruction time was increased by approximately 50% [16]. Kertész et al. reported a reconstruction time of up to 22 h for their parallel implementation [20]. Additionally, they incorporated the model in both the forward and backward projection, and they reported an image noise reduction of 11% [20]. However, Cal-González et al. observed enhancement of the image noise and, thus, increased the number of image updates of the reconstruction without correction by a factor of 2.6, achieving the same image noise levels between reconstructions with and without correction [16]. A summary of the reconstruction-based positron range correction techniques is given in Table 2.

#### 3.1.2. Post-Reconstruction Approaches

Post-reconstruction approaches are correction techniques that are applied to the reconstructed PET image and, as such, are not part of the reconstruction algorithm itself. Rukiah et al. [21] proposed a positron range correction for ^68^Ga based on the Richardson–Lucy (RL) de-blurring method [24,25]. Monte Carlo simulations were performed to obtain the positron range distribution profiles of ^68^Ga for different homogeneous tissues from which kernels were created [21]. Simulations of phantom measurements were performed for both ^68^Ga and ^18^F. Rukiah et al. reported that spatial resolution of the corrected ^68^Ga PET images was inferior to ^18^F PET images by approximately 9%, 1%, and 56% for water, bone, and lung tissue, respectively [21]. Additionally, noise was enhanced in the corrected ^68^Ga PET images compared to the uncorrected ^68^Ga PET images by 159% in lung tissue, while for bone tissue and water, the noise increased by 13% and 17%, respectively [21].

Deep learning techniques were also used to correct for positron range effects. Herraiz et al. implemented a positron range correction based on convolutional neural networks (CNNs) [22]. They generated ^68^Ga and ^18^F PET images of mice models using Monte Carlo simulations. The ^18^F PET images were used as a reference. Data augmentation was performed to increase the amount and variety of training data. The corrected ^68^Ga PET images generated by this deep learning-based positron range correction method demonstrated increased RC up to approximately 80% and lower noise levels up to approximately 37% in comparison to the uncorrected ^68^Ga PET images [22]. Compared to the ^18^F PET images, the corrected ^68^Ga PET images could achieve comparable RC and noise levels [22]. When implemented on a graphics processing unit (GPU), the required time for positron range correction was only a few seconds [22]. Yang also implemented a deep learning correction method for ^68^Ga utilizing networks with different architectures [23]. For this study, Monte Carlo simulations of phantoms were performed using ^68^Ga and another source of gamma rays with no positron range effect. The CNNs were trained using the ^68^Ga PET images as inputs and the images produced from the gamma source as a reference. Yang reported that the corrected images had sharper boundaries, higher RC, and a higher spillover ratio (SOR) [23].

### 3.2. Noise Reduction Techniques

Two different approaches were identified for reducing the noise in ^68^Ga PET images and enabling low-count ^68^Ga PET measurements. The first approach reduces the noise during the image reconstruction process (reconstruction-based noise reduction approaches, *n* = 11) [26,27,28,29,30,31,32,33,34,35,36], whereas the second approach is based on neural networks (deep learning approaches for noise reduction, *n* = 5) [37,38,39,40,41], as seen in Figure 2.

#### 3.2.1. Reconstruction-Based Noise Reduction Approaches

Ordered subset expectation maximization (OSEM) is a reconstruction algorithm that is generally used to reconstruct PET images [42]. It is an iterative expectation maximization (EM) algorithm that tries to find the maximum likelihood (ML) solution, which corresponds to the most likely image given the measured projection data. During each iteration, a subset of the measured data is employed to estimate a new image with a higher likelihood. However, with each update of OSEM, the noise in the reconstructed image increases because of the noisy projection data. Therefore, one approach is to limit the number of updates during reconstruction limiting the reconstructed image noise. However, this means that full convergence is not guaranteed, which may result in an underestimation of radioactivity concentrations [43,44]. Another approach is to ensure full convergence of the reconstructed image by applying enough updates during the reconstruction, while a low-pass filter is applied to the final reconstructed image to reduce the impact of noise.

An approach that allows full convergence is the Bayesian penalized likelihood (BPL) or regularized iterative reconstruction algorithm. In comparison to a regular OSEM algorithm, a noise regularization or penalty term is added to the objective function with a weighting or beta value (β value). Higher β values depict a higher weighting of the regularization term resulting in a more effective noise suppression during the reconstruction. However, including a penalty term in the objective function generally needs an adaption of the optimization scheme as well, with Block-Sequential Regularized Expectation Maximization (BSREM), an optimizer that is frequently used for BPL reconstruction algorithms [45,46]. Based on the studied literature two BPL reconstruction algorithms have been investigated with respect to ^68^Ga-labeled radiotracers.

One of those BPL reconstruction algorithms was introduced by GE Healthcare under the name Q.Clear, which includes a noise regularization term based on relative differences between neighboring voxels and uses BSREM as an optimizer. In addition, Q.Clear incorporates both time-of-flight (TOF) information and point spread function (PSF) modeling. A literature search including both BSREM and Q.Clear revealed multiple studies on the optimal use of Q.Clear for different ^68^Ga-labeled radiotracers using both patient and phantom studies. These studies demonstrated that increasing the β value resulted in a reduction in the image noise [26,27,28,29,30,31,32,33], an improvement in the signal-to-noise ratio (SNR) [26,27,29,30,32], and a reduction in Gibbs artifacts [33] at the cost of a reduction in the maximum standardized uptake value (SUV_max_) [26,28,29,31], CR [26,28,30], RC [33], and signal-to-background ratio (SBR) [27,29,32]. Additionally, the mean standardized uptake value (SUV_mean_) was almost identical for different β values [26,27,28,29]. Increasing the number of iterations had a minor effect on image noise in BSREM [26]. It was further observed that the same β values have different impacts on SUV_max_, CR, and SNR of lesions with different sizes and uptakes. Specifically, an increase in the β value enhanced the relative difference in SUV_max_, CR, and SNR as the size and uptake of lesions decreased [26,29,32].

Quantitative image analysis and visual assessment by experts confirmed that BSREM can outperform OSEM with respect to image quality. However, these findings were only valid for a certain range of β values. Different studies investigated the potential of BSREM in comparison to OSEM for different ^68^Ga-labeled radiotracers, while using different imaging settings and measures of image quality. As a result, different ranges of recommended β values were reported, as presented in Table 3. BMI can also affect the optimal β value as Zanoni et al. suggested a rather large β value of 1600 for ^68^Ga-DOTANOC PET scanning of scan patients with 25 ≤ BMI < 30 or BMI ≥ 30 [34]. However, these ranges of β values need to be interpreted with care since this is a GE proprietary and unitless tuning parameter, which can also depend on the version of the reconstruction software. Nevertheless, the different ranges of optimal β values clearly demonstrate optimal noise regularization is highly dependent on the acquisition time, injected dose, and ^68^Ga-labeled radiotracer.

Additionally, studies demonstrated the potential to reduce the acquisition time using BSREM while maintaining an image quality that is still comparable to OSEM reconstructions using longer acquisition times [27,29,32]. Based on quantitative image analysis, up to a 75% reduction in acquisition time was achieved [27], while another study reported a potential acquisition time reduction of up to 33% based on a visual assessment [32]. Additionally, Svirydenka et al. demonstrated that the use of BSREM and a five-fold increase in the acquisition time resulted in a ten-fold reduction in the administered activity of ^68^Ga-PSMA in comparison to standard OSEM [31], therefore enabling ultra-low activity examinations.

In addition to Q.Clear, United Imaging Healthcare introduces a BPL iterative reconstruction algorithm based on total variation regularized expectation maximization TVREM (HYPER Iterative) with a penalization factor between 0 and 1 to adjust the total variation penalization of voxels of corresponding neighborhoods. TVREM was used for PET phantom and patient scanning with ^68^Ga-PSMA (20 patients) [35] and ^68^Ga-DOTATATE (17 patients) [36]. As the penalization factor α increased, image noise [35,36], CR [35], SUV_max_ [35,36], and TBR [36] were reduced, while SNR increased [36]. SUV_mean_ remained similar when the penalization factor increased [35,36]. Additionally, it was observed that the penalization factor α had a greater effect as the size of the lesions decreased. Specifically, for lesions with a diameter ranging between 10 and 20 mm, the SUV_max_ was decreased as the penalization factor α increased, while for lesions with a diameter greater than 20 mm, the SUV_max_ was almost identical [35]. In addition to that, the relative differences in SNR and TBR were larger for lesions with a diameter lower than 10 mm than lesions with greater or equal [36].

Quantitative image analysis and visual assessment by experts demonstrated that TVREM was capable of enhancing image quality more compared to OSEM but only for a certain range of values [35,36]. Yang et al. demonstrated that for penalization factor α ranging between 0.07 and 0.28 for ^68^Ga-PSMA, a reduction in the acquisition time of 33% was achieved based on quantitative image analysis and visual assessment from experts [35]. Specifically, compared to OSEM, they reported improvements in the contrast up to 17%, the SUV_max_ up to 15%, and the image noise was reduced up to 32% [35]. Liu et al. reported that TVREM improved the overall SUV_max_, SNR, and TBR compared to OSEM [36]. Additionally, based on visual assessment, they suggested that TVREM was capable of preserving the image quality for ^68^Ga-DOTATATE while achieving a reduction in the acquisition time by 33% for values between 0.14 and 0.35 [36].

#### 3.2.2. Deep Learning Approaches for Noise Reduction

The potential of deep learning approaches, both supervised and unsupervised, to reduce the noise in PET images has been demonstrated in the literature. For a supervised approach, low- and high-count PET images are available for each scan to train the network. Liu et al. implemented a supervised deep learning approach in which cross-tracer and cross-protocol learning were performed [37]. They used PET images consisting of only 10% of the counts of the original single-bed ^18^F-FMISO (*n* = 12), single-bed ^18^F-FDG (*n* = 9), whole-body ^18^F-FDG (*n* = 12), and whole-body ^68^Ga-DOTATATE (*n* = 15) PET scans. Image quality enhancement was reported by comparison with the unenhanced, low-count PET images. Additionally, they reported the potential of employing neural networks trained based on ^18^F-FDG image data to reduce the image noise for other radiotracers such as ^68^Ga-DOTATATE and different scanning protocols [37]. Deng et al. used low-count ^68^Ga-PSMA PET images (41 patients) and data augmentation to train a supervised deep learning model for denoising [38]. Based on quantitative analysis and visual assessment, Deng et al. suggested a 50% count reduction as the optimal trade-off.

Additionally, unsupervised approaches were also investigated in the literature. In comparison to supervised approaches, training pairs are not required to train the unsupervised approaches, since there is no target output. Therefore, high-count PET images are unnecessary to train an unsupervised deep learning approach. Cui et al. investigated the potential of unsupervised deep learning [39,40]. An unsupervised neural network was validated using simulated ^18^F-FDG phantom data and evaluated using ^68^Ga-PRGD2 PET/CT (*n* = 10) and ^18^F-FDG PET/MRI patient data (*n* = 30) [40]. Compared to the original PET images, the mean CNR was improved by approximately 53% for ^68^Ga-PRGD2 PET/CT and approximately 47% for ^18^F-FDG PET/MRI [40]. However, they suggested radiotracer- and modality-independent effects since they reported no statistically significant differences between those improvements [40]. In a later publication [41], Cui et al. expanded their previous approach [40] by incorporating a second unsupervised neural network, which used, as initial parameters, the pre-trained parameters of the first network. In that study [41], the same dataset was used as in [40]. This new approach improved the mean CNR by approximately 71% for ^68^Ga-PRGD2 PET/CT and by approximately 58% for ^18^F-FDG PET/MRI compared to the original PET images. In addition, compared to the previous approach [40], it could better retain better structural image features [41]. Meanwhile, SUV_max_ and SUV_mean_ values were comparable for both approaches [41].

## 4. Discussion

In the studied literature, the capability of improving the image quality of ^68^Ga was demonstrated by incorporating a ^68^Ga-specific positron range correction. In some cases, image quality comparable to ^18^F was reported [21,22]. The introduction of a ^68^Ga-specific positron range correction may increase the ^68^Ga image quality in clinical practice and improve the diagnostic value of these images. Despite that, none of these techniques have yet been applied or validated in a clinical setting. To the best of our knowledge, from the presented studies, only the techniques presented in [16] were applied in a preclinical setting (18 scans of 12 mice), where an improvement in the image quality in relation to the case of no correction was reported [47]. However, there is the need for ^68^Ga-specific positron range correction to be translated into the clinical setting after which thorough validation is required prior to clinical implementation.

Another aspect of the ^68^Ga-specific positron range corrections that needs to be considered for clinical applications is the computation time. Advanced and accurate reconstruction-based corrections have increased the computation time [16,20], which may be incompatible with clinical practice. Therefore, for the reconstruction-based corrections, one necessary criterion that needs to be fulfilled is to have a minor impact on the overall computational time.

The post-reconstruction corrections also showed promising results. Deep learning approaches have the potential to improve image quality while maintaining low noise levels. Aside from that, they seem to be computationally time efficient. It was demonstrated that a post-reconstruction deep learning correction implemented in GPU required only a few seconds [22]. However, for deep learning approaches, large datasets are required, including as many different cases as possible and reducing the potential of overfitting. Data augmentation techniques can increase the size of the dataset and reduce overfitting [48] and have already been applied [22]. However, for small datasets without enough representative training data, data augmentation techniques will not be able to correct the lack of insufficient data.

Noise is another important factor that diminishes image quality. Iterative reconstruction algorithms using penalizing factors have the potential to produce better image quality than OSEM and even reduce acquisition time. However, quantitative image analysis demonstrated a greater reduction in acquisition time in comparison to the reduction suggested through visual assessment. Additionally, the intrareader variability may be considered when performing a visual assessment.

In contrast to OSEM, a small effect on image noise was reported in Q.Clear as the number of iterations increased [26]. This will allow for better conversion of small structures in particular, which will enhance small lesion identification and quantification [49]. Increasing the β value resulted in higher relative differences in SUV_max_, CR, and SNR for small lesions, while as the size and uptake increased, this effect became less prominent [26,29,32]. Q.Clear can enhance small lesion identification and quantification; however, for too-large β values, the noise suppression effect of Q.Clear may hinder the detection of small lesions in comparison to OSEM. Additionally, identifying the optimal values of the noise penalization factor where the generated images are better than OSEM is not a straightforward process. The optimal value of the noise penalization factor depends on various factors such as the radiotracer distribution, acquisition time, BMI, clinical indication, measures of image quality, and even the preference of the specialists or experts evaluating the images. Consequently, studies usually reported a range of recommended values. However, ideally, the noise penalization factor should be evaluated for each clinical application.

Deep learning techniques were also implemented to denoise data. In the literature, both supervised and unsupervised approaches were described. However, studies did not only explicitly focus on ^68^Ga-labeled radiotracers but also a combination of ^68^Ga-labeled and ^18^F-labeled radiotracers. It was demonstrated that supervised deep learning enabled networks trained from one radiotracer or scanning protocol to denoise PET images of other radiotracer or scanning protocols [37]. This was demonstrated using PET images consisting of only 10% of the counts of full-activity PET images [37]. However, Deng et al. showed that PET images generated from 50% of the counts using a deep learning approach revealed the same clinical result as the original data in patients with a suspected diagnosis of prostate cancer based on a clinical quantitative assessment by specialists [38]. Unsupervised deep learning approaches also demonstrated the potential of denoising. One advantage is the absence of high-quality PET images as training pairs. However, none of the included studies compared the image quality produced by unsupervised learning with respect to supervised learning.

One disadvantage of the reconstruction-based noise reduction approaches in comparison to the deep learning approaches is the need to define the optimal value of the noise penalization factor, which depends on various factors that are highly variable. Furthermore, it was demonstrated that using supervised deep learning networks can be trained from one radiotracer or scanning protocol and used for denoising other radiotracer or scanning protocols [37]. Despite that, one important disadvantage of deep learning approaches is the dependency on the training dataset. Deng et al. performed data augmentation to increase the dataset size and avoid overfitting [38]. However, data augmentation techniques will not be able to compensate if there are not sufficient representative training data.

In the studied literature, either positron range corrections or denoising approaches were investigated with respect to ^68^Ga. Ideally, in a clinical setting using ^68^Ga-labeled radiotracers, there is a need for both positron range correction and noise reduction techniques. The ^68^Ga PET image quality can be enhanced by implementing positron range correction and can even achieve comparable image quality as ^18^F PET. However, some positron range correction techniques were reported to also enhance image noise. Therefore, combining a denoising approach that will enable low-count PET imaging may compensate for the increased image noise introduced by the positron range correction. Low-count PET imaging can be translated into lower administered activities and/or acquisition time, which has the potential to reduce the cost of PET imaging, increase the patient throughput of the system, and increase patient comfort.

## 5. Conclusions

Positron range correction techniques have the potential to improve the ^68^Ga PET image quality, and it has been shown that image quality comparable to ^18^F can be achieved. However, none of the proposed approaches have been clinically evaluated. Therefore, prior to clinical implementation, thorough validation is necessary.

Different denoising approaches for ^68^Ga that were evaluated in both phantom and clinical data, based on both reconstruction and deep learning techniques, showed the potential to achieve a substantial reduction in administered radiotracer activity or acquisition time.

The combination of both positron range correction and image denoising techniques has great potential to significantly improve the ^68^Ga PET image quality and consequently enhance the clinical value of low-count PET using ^68^Ga-based radiotracers.

## Figures and Tables

**Figure 1 diagnostics-12-02335-f001:**
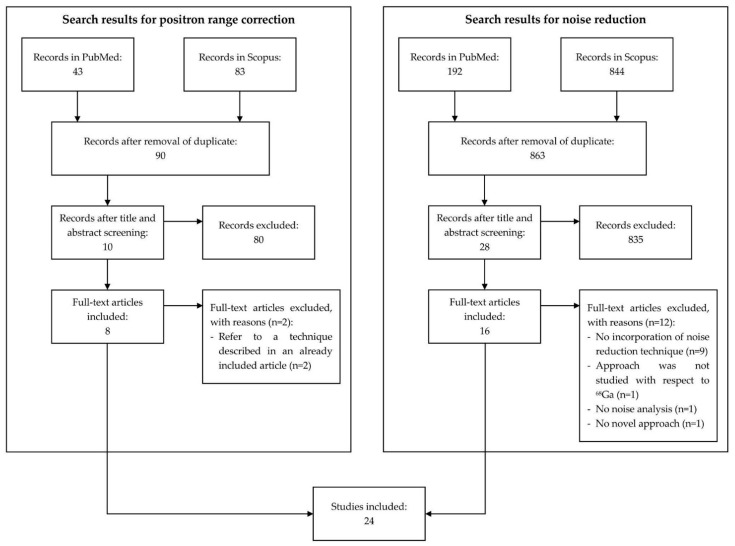
Chart illustrating the screening process of the literature review.

**Figure 2 diagnostics-12-02335-f002:**
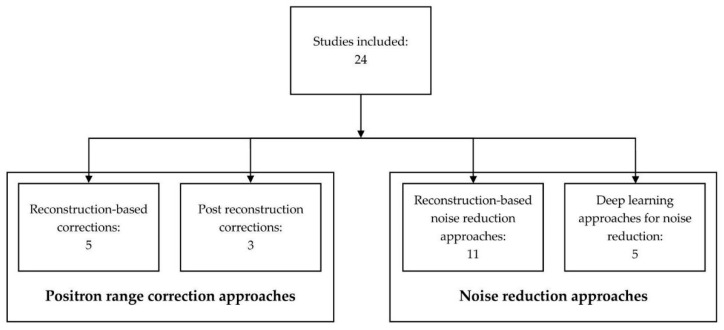
Categorization of the literature search with the numbers depicting the number of articles per category.

**Table 1 diagnostics-12-02335-t001:** Summary of abbreviations in this paper.

Abbreviation	Description
^18^F	Fluorine-18
^68^Ga	Gallium-68
^68^Ge	Germanium-68
BMI	Body mass index
BPL	Bayesian penalized likelihood
BSREM	Block-sequential regularized expectation maximization
CNN	Convolutional neural network
CNR	Contrast-to-noise ratio
CR	Contrast recovery
CT	Computerized tomography
EM	Expectation maximization
*E* _max_	Maximum energy
FWHM	Full width half maximum
FWTM	Full width tenth maximum
GMP	Good manufacturing practice
GPU	Graphics processing unit
min/bp	Minutes per bed position
ML	Maximum likelihood
MRI	Magnetic resonance Imaging
OSEM	Ordered subset expectation maximization
PET	Positron emission tomography
PSMA	Prostate-specific membrane antigen
RC	Recovery coefficients
SBR	Signal-to-background ratio
SNR	Signal-to-noise ratio
SOR	Spill-over ratio
SUV_max_	Maximum standardized uptake value
SUV_mean_	Mean standardized uptake value
TBR	Tumor-to-background ratio
TOF	Time-of-flight
TVREM	Total variation regularized expectation maximization

**Table 2 diagnostics-12-02335-t002:** Summary of the reconstruction-based corrections for ^68^Ga. The asterisk (*) depicts an improvement observed by incorporating the blurring kernel both in forward and backward projection steps.

Positron Range Correction	Reported Improvements	Reported Drawbacks
Tissue-independent	Improve FWTM [16] Improve activity recovery [17,18]	FWTM and activity recovery over-correction in the bone tissue [16,18] FWTM under-correction in lung tissue [16] Increased noise [17]
Homogeneous tissue-dependent	Improved FWTM [16] and FWHM [18] Improved TBR [16] Improve activity recovery [17,18] Improved RC [18] Improved contrast [19] Reduced misplaced activity concentration in soft tissue [19]	Increased reconstruction time [16] Increased noise [16,17] Gibbs artifacts [16,17]
Heterogeneous tissue-dependent	Improved FWTM [16] Improved TBR [16] Improved contrast [19] Reduced misplaced activity concentration in soft tissue [19] Improved RC [20] Improved CR [20] Improved CNR [20] Reduced image noise * [20]	Increased reconstruction time [16,20] Increased noise [16] Gibbs artifacts [16,20]

**Table 3 diagnostics-12-02335-t003:** Summary of the recommended β values based on the findings of studied literature. Those values were reported to have better overall performance than OSEM. (*) The respective studies did not report the median or mean injected dose. (**) The recommended β value for overweight patients as was suggested by the corresponding authors.

Recommended β Values	Radiotracer	Number of Patients	Acquisition Time	Injected Dose	References
400–550	^68^Ga-PSMA	25	2 min/bp	128 MBq (median)	[26]
400–900	^68^Ga-PSMA	20	2 min/bp	2 MBq/kg (mean)	[27]
800–1000	^68^Ga-PSMA	20	1 min/bp	2 MBq/kg (mean)	[27]
1200–1400	^68^Ga-PSMA	20	0.5 min/bp	2 MBq/kg (mean)	[27]
500–750	^68^Ga-PSMA	36	4 min/bp	151.7 MBq (median)	[28]
550	^68^Ga-PSMA	25	15 min, single bed position	15 MBq *	[31]
500–750	^68^Ga-RM2	42	4 min/bp	144.3 MBq (median)	[28]
500	^68^Ga-Citrate	6	4–8 min/bp	242.35 MBq (mean)	[30]
400	^68^Ga-DOTATOC	13	2 min/bp	2.3 MBq/kg (mean)	[29]
533	^68^Ga-DOTATOC	13	1.5 min/bp	2.3 MBq/kg (mean)	[29]
1100–1400	^68^Ga-DOTATATE	30	1.5 min/bp	160.3 MBq (mean)	[32]
1300–1600	^68^Ga-DOTATATE	30	1 min/bp	160.3 MBq (mean)	[32]
1600 **	^68^Ga-DOTANOC	75	3 min/bp	100–200 MBq *	[34]

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
