# Peer review of "Positron Range Corrections and Denoising Techniques for Gallium-68 PET Imaging: A Literature Review"

_diagnostics, 2022, doi:10.3390/diagnostics12102335_

Round 1

Reviewer 1 Report

The paper comprises a thorough literature review of the topic of PET imaging denoising. 

My only comment is that the paper does not clearly state that this is essentially a literature review, which should be explicit in the abstract and introduction.

Author Response

Based on the comment of the reviewer, we have changed the title to clearly state that it is a literature review. In addition, we have made some changes to the abstract and introduction and explicitly mention that it is a literature review, as it was suggested by the reviewer 1.

Reviewer 2 Report

The work is very interesting. Written in an understandable and clear way. I recommend approval as it is.

Author Response

  • The second reviewer was satisfied with the submitted version of the paper without any changes.

Reviewer 3 Report

The paper is well written and useful to investigate image quality in a clinical setting.

Author Response

  • The third reviewer was satisfied with the submitted version of the paper without any changes.

Reviewer 4 Report

Contributions:

This study reviews positron range correction techniques for 68Ga and noise reduction methods to enhance the image quality of low count 68Ga PET imaging. My comments are given below:

1. The depth of this paper is not sufficient.

A.The first three and the last authors’ paper was not cited in the reference. If the authors did not publish journal papers, it is not adequate to write a review paper.

B.Only Prof. Mottagphy’s papers have been cited in [2][7]. However, Prof. Mottagphy is the 12th and 8th author. The contributions are limited.

C.(Page 9) Although the authors have compared many methods, these methods did not been deeply studied.   

D.Only 49 papers have been cited; it is not sufficient for a review paper. The scope is limited. In addition, only a reference paper [16] comes from IEEE Transactions. So the review scope is not sufficient.

2. Figure 1 is blurred. 

3. The blocks for full-text articles excluded with reasons are too redundant in Fig. 1.

4. (Line93 on page 3) The coding term n should be defined.

5. (Page 4) The sub-section 3.1.1 describes the reconstruction-based correction. However, no papers have been cited.

6. (Page 6)The captions for Tables 1 and 2 are too redundant.

7. There are too many abbreviations in this paper. Please create an abbreviation table.

8. The English written should be proofread by a native speaker. Line 249 on page 7 should be revised.

9. (Page 8)The sub-grid lines in Table 2 should be removed.

10. (Line 307 on page 9) A reference should be cited.

Author Response

1A and 1B:  “The first three and the last authors’ paper was not cited in the reference. If the authors did not publish journal papers, it is not adequate to write a review paper.” and “Only Prof. Mottagphy’s papers have been cited in [2][7]. However, Prof. Mottagphy is the 12th and 8th author. The contributions are limited.

Although the authors do not have publications of own original work on the topics investigated in this study, they are experts in the field of PET imaging and carefully considered the included studies. Additionally, our research is objective since there is no bias from the authors and considering the very positive feedback from the other 3 referees, we feel that the quality of our manuscript is high. 

1C:  “(Page 9) Although the authors have compared many methods, these methods did not been deeply studied.

Although we understand the motivation of this comment, it must be said that going deeper into details is beyond the scope of this literature review paper, as we want to inform the readers about the different available approaches and not dive in the technical details. We feel that providing more technical details would negatively affect the readability of the paper. Readers who are interested in the exact technical details of each technique can find these in the original papers cited in the review article.

1D: “Only 49 papers have been cited; it is not sufficient for a review paper. The scope is limited. In addition, only a reference paper [16] comes from IEEE Transactions. So, the review scope is not sufficient.

We respectfully disagree with the reviewer’s opinion that a minimum number of cited references is required for producing a review paper. The number of the cited papers depends strongly on the amount of research that has been performed on that topic. As described in the paper, we created search queries for this topic in both Scopus and PubMed to ensure all relevant publications were included. Therefore, we are convinced we have included all relevant papers in our review article. If to the opinion of the referee any specific relevant publications are still missing, we would gladly add them upon indication. Aside from that, there are also other published review articles that had similar number of papers cited such as: https://doi.org/10.1007/s00259-017-3775-4. Furthermore, we have cited 6 in total papers from IEEE. Specifically, there are 4 papers [16, 42, 45, 46] from IEEE Transactions on Medical Imaging, a paper [19] from IEEE Transactions on Nuclear Science and a paper [21] from IEEE Nuclear Science Symposium and Medical Imaging Conference Proceedings. Additionally, it must be mentioned that we cannot include irrelevant papers from IEEE Transactions.

2:    “Figure 1 is blurred.

We fixed the mentioned figure and we changed a bit the size of it which was also degrading the quality of the figure.

3:    “The blocks for full-text articles excluded with reasons are too redundant in Fig. 1.

We agree that the mentioned blocks are redundant as this information is also provided in the main text; however, we feel this information greatly helps the reader to better understand the figure 1.

4:    “(Line93 on page 3) The coding term n should be defined.

We believe that the term n is common knowledge in scientific publications and therefore understandable without further definition, especially if we consider the figure 1 which is on the previous page and describes the screening process.

5:    “(Page 4) The sub-section 3.1.1 describes the reconstruction-based correction. However, no papers have been cited.

We have cited the relevant papers as suggested by the referee.

6:    “(Page 6) The captions for Tables 1 and 2 are too redundant.

We removed some redundant parts of those captions as suggested by the referee.

7:    “There are too many abbreviations in this paper. Please create an abbreviation table.

As suggested by the referee we added a new table (Table 1) in the introduction that provides a description for the abbreviations that are being used.

8:    “Line 249 on page 7 should be revised.

We revised that part.

9:    “(Page 8) The sub-grid lines in Table 2 should be removed.

We removed the sub-grid lines as it was suggested.

10:  “(Line 307 on page 9) A reference should be cited.

We added the citation as it was suggested.

Round 2

Reviewer 4 Report

This paper is a literature review. In my previous comment 4, I think n should be defined. 

Author Response

We defined the term n as suggested by the referee.